# SID-TGAN: A Transformer-Based Generative Adversarial Network for Sonar Image Despeckling

**Xin Zhou** [1],[†] , **Kun Tian** [1],[*],[†] , **Zihan Zhou** [1] , **Bo Ning** [1] and **Yanhao Wang** [2]

1 School of Information Science and Technology, Dalian Maritime University, Dalian 116026, China; zhouxin314159@dlmu.edu.cn (X.Z.); zmzzh@dlmu.edu.cn (Z.Z.); ningbo@dlmu.edu.cn (B.N.)
2 School of Data Science and Engineering, East China Normal University, Shanghai 200062, China; yhwang@dase.ecnu.edu.cn
* Correspondence: tkflying@dlmu.edu.cn
† These authors contributed equally to this work.

**Abstract:** Sonar images are inherently affected by speckle noise, which degrades image quality and hinders image exploitation. Despeckling is an important pre-processing task that aims to remove such noise so as to improve the accuracy of analysis tasks on sonar images. In this paper, we propose a novel transformer-based generative adversarial network named SID-TGAN for sonar image despeckling. In the SID-TGAN framework, transformer and convolutional blocks are used to extract global and local features, which are further integrated into the generator and discriminator networks for feature fusion and enhancement. By leveraging adversarial training, SID-TGAN learns more comprehensive representations of sonar images and shows outstanding performance in speckle denoising. Meanwhile, SID-TGAN introduces a new adversarial loss function that combines image content, local texture style, and global similarity to reduce image distortion and information loss during training. Finally, we compare SID-TGAN with state-of-the-art despeckling methods on one image dataset with synthetic optical noise and four real sonar image datasets. The results show that it achieves significantly better despeckling performance than existing methods on all five datasets.

**Keywords:** sonar image; speckle denoising; generative adversarial network (GAN); transformer

## 1. Introduction

With the rapid development of ocean engineering, underwater sonar imaging systems have been successfully applied to many different tasks, including underwater target detection and tracking, marine environment monitoring, and seabed resource exploration. However, sonar imaging systems are often disturbed by various kinds of noise during the imaging process due to the complexity and diversity of the underwater environment, resulting in poor visualization, low resolution, weak picture texture, and blurred edges of sonar images [1]. In particular, speckle noise caused by bottom reverberation, which can come from the sea surface or surrounding scatterers, seawater and marine life scatterers, seabed or surrounding scatterers [2], is a common type of noise that seriously affects the quality of underwater sonar images. To better extract and interpret information from images and thus further improve the accuracy of downstream analysis tasks, denoising speckle noise (or *despeckling*) has been regarded as an essential preprocessing step for sonar image processing.

The interest in sonar image despeckling has been intense since the early 1980s. From then on, many effective filtering-based methods [3–9] were proposed for sonar image despeckling. However, filtering-based methods suffer from two fundamental drawbacks. First, they require prior statistical knowledge of speckle noise in sonar images to effectively reduce it. However, such knowledge is difficult to obtain in advance. Second, they inevitably lead to losses in the resolution, textures, and many other details of sonar images after denoising. Therefore, more recently, researchers have turned their attention to deep

neural network-based methods [10–17] for sonar image despeckling. In particular, convolutional neural networks (CNNs) are the most widely adopted due to their strong abilities for spatial feature extraction, which is important to reduce speckle noise from sonar images since such noise is inherently spatial-aware. As such, CNN-based methods have achieved much better despeckling performance than filtering-based methods in the sense that they do not require any prior statistical knowledge and cause much fewer losses of image quality. However, since CNNs typically only consider the relationships between neighboring pixels, they cannot capture global features of images, which are potentially used jointly with local (spatial) features to further improve the performance of sonar image despeckling.

To overcome the limitation of CNN-based methods for despeckling by extracting global characteristics and combining them with local (spatial) characteristics, we propose a Transformer-based Generative Adversarial Network for Sonar Image Despeckling named SID-TGAN in this paper. Our main contributions are summarized as follows:

- We design the architecture of SID-TGAN with transformer and convolutional blocks to perform a multi-scale local-to-global feature learning so as to extract more comprehensive features from the input images, which are then integrated into the generator and discriminator networks for feature fusion and enhancement.
- We propose a new adversarial loss function for SID-TGAN, which combines image content, local texture style, and global similarity to better preserve the mapping of the relationship between local and global information and reduce image distortion and information loss during training.
- We conducted extensive experiments to compare SID-TGAN with several state-of-the-art despeckling methods on an image dataset with synthetic optical noise and four real sonar image datasets. The results show that SID-TGAN has substantially better despeckling performance by achieving 2.53–39.84%, 19.23–48.36%, 2.50–32.20%, 17.00–61.03%, and 11.32–15.43% improvements in image quality than existing methods in the synthetic and real noise settings.

**Paper Organization**. The rest of this paper is organized as follows. We discuss the relevant work to sonar image denoising in Section 2. Then, we introduce our SID-TGAN model in detail in Section 3. We present the experimental results to verify the despeckling performance of SID-TGAN in Section 4. Finally, we conclude this paper and indicate possible directions for future work in Section 5.

## 2. Related Work

### 2.1. Filtering-Based Methods for Speckle Denoising

The classic methods to remove speckle noise from images are based on filtering, generally divided into local filtering and non-local filtering. Local filtering methods, such as median and mean filtering, are based on adjusting the filtering window according to local image statistics to deal with signal nonstationarity. They only achieve good noise removal performance in a uniform area but lead to blurred boundaries when applied to a non-uniform area. To address this issue, non-local filtering methods were proposed to better preserve the edge information of images. Danielyan et al. [4] proposed the Block Matching 3D (BM3D) algorithm, where the denoising and deblurring operations are decoupled based on the Nash equilibrium balance of their objective functions. BM3D was known to achieve good speckle denoising performance on sonar images. Han et al. [18] proposed an improved BM3D sonar image denoising algorithm, which adjusts parameters based on the noise characteristics of sonar images and incorporates Gaussian filtering and grayscale correction before the basic estimation to eliminate speckle noise. Fan et al. [5] considered converting multiplicative speckle noise into additive Gaussian noise and performing speckle denoising on sonar images through adaptive BM3D. Chen et al. [8] proposed a speckle noise reduction method for side-scan sonar images based on adaptive BM3D. Wang et al. [7] proposed an adaptive non-local spatial information denoising method with improved performance on underwater sonar images. Furthermore, Wang et al. [9] proposed a shearlet transform-based noise reduction method for sonar images. Jin et al. [6] used a saliency detection method

based on manifold sorting for sonar image processing, automatically segmenting a sonar image into salient and non-salient areas. Li et al. [19] proposed a denoising method for sub-bottom profile sonar images that combines guiding weights with a non-local low-rank filtering framework. Chaillan et al. [20] proposed an adaptive approach that combines a multi-resolution transformation and a filtering method to reduce speckle noise in synthetic aperture sonar (SAS) images [21,22].

Although the above filtering-based speckle denoising methods can achieve good denoising performance in some cases, they require accurate noise estimation to obtain prior statistical knowledge of speckle noise from sonar images for effective denoising. However, accurately estimating speckle noise is very difficult on real images. In addition, they are prone to damaging the details and textures of sonar images, which reduces their quality. The above shortcomings limit their usage in practical scenarios.

### 2.2. Deep Learning Methods for Speckle Denoising

Convolutional neural networks (CNNs), which have achieved great success in different image-processing tasks, have also been applied to image denoising due to their powerful spatial feature extraction and representation capabilities. Chierchia et al. [10] proposed a CNN-based image despeckling method called SAR-CNN, which generates clean images through multi-temporal fusion and uses residual learning and discriminative model learning for denoising. An auto-encoder model that could reduce speckle noise from sonar images while maintaining their resolutions was proposed in [11]. They then used a loss function based on structural similarity measures to reduce speckle noise while preserving important geometric properties of sonar images. Liu et al. [23] introduced a hybrid denoising approach that employs CNN and consistent cycle spinning (CCS) in the nonsubsampled shearlet transform (NSST) domain for synthetic aperture radar (SAR) images. Huang et al. [13] proposed the Neighbor2Neighbor method for image denoising. Neighbor2Neighbor used a random neighborhood subsampler to generate training image pairs, trained a denoising network on the generated subsampled training pairs, and introduced a regularization term into the loss function to obtain better performance. By avoiding heavy dependence on the noise distribution assumptions, it has good denoising performance on images in various domains, including sonar images. Vishwakarma [17] incorporated sparse representation techniques into CNNs to improve the denoising and inpainting of sonar images. Chen et al. [14] proposed an ANLResNet model for sonar image despeckling by combining SRResNet with non-local blocks of asymmetrical pyramids for speckle noise in sonar images. Zhou et al. [15] proposed a self-supervised denoising method for sonar images without high-quality reference images since obtaining such references during the sonar image denoising process was often difficult. Perera et al. [24] proposed a transformer-based model for sonar image despeckling.

The above despeckling methods for sonar images are primarily based on CNNs. However, CNNs cannot directly model the global context because they only use local receptive fields for feature extraction. Transformer blocks can extract the dependencies between distant pixels and parallelize hierarchical representation learning to extract global features of images. However, the local and global features extracted by the convolutional and transformer blocks are still poorly integrated using the existing model architectures. To address the above issues, we are inspired by [25] to propose a transformer-based generative adversarial network for sonar image despeckling named SID-TGAN.

## 3. The SID-TGAN Model

### 3.1. Problem Statement

We consider that sonar images are affected by multiplicative speckle noise during imaging. The relationship between an observed noisy image $Y$ with multiplicative speckle noise $N$ and a clean image $X$ is $Y = XN$. We do not make any assumption about the distribution of $N$. The goal of despeckling is to remove the speckle noise $N$ from a noisy image $Y$ and recover the clean image $X$.

### 3.2. Model Architecture

Similar to the original generative adversarial network (GAN), our proposed SID-TGAN model consists of two components, namely the generator and discriminator networks. For a noisy input image $Y$, the goal of the generator network is to produce a despeckled image $Y_G$. Then, the goal of the discriminator network is to distinguish the produced image $Y_G$ from the clean image $X$, thus improving the quality of the despeckled images through adversarial training. Next, we will present the architectures of the generator and discriminator networks, respectively, and introduce the adversarial loss function that we use for training.

#### 3.2.1. The Generator Network

The architecture of the generator network in SID-TGAN is shown in Figure 1.

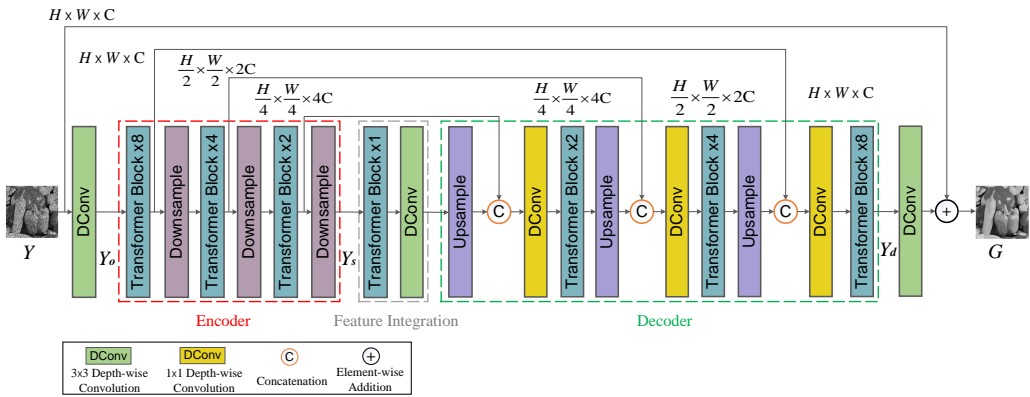

**Figure 1.** The architecture of the generator network in SID-TGAN.

The three key components to produce the despeckled image $Y_G$ for a noisy input image $Y$ in the generator network are as follows:

- **Encoder:** It first obtains the low-level shallow features $Y_o$ of the noisy image $Y$ through a deep convolutional layer with a kernel size of $3 \times 3$. Then, it acquires the high-level features $Y_s$ from $Y_o$ through an encoder. The encoder consists of three transformer layers and three down-sampling operations to capture the context in the image $Y$. The first layer of the encoder consists of eight transformer blocks, with the number of subsequent transformer blocks decreasing by half per layer. The spatial size of the feature map in each encoder layer is reduced half layer by layer through the down-sampling operation, while its channel capacity is doubled accordingly.

- **Feature Integration:** The component for feature integration consists of a transformer layer with two transformer blocks and a convolutional layer with a kernel size of $3 \times 3$. It connects the encoder and the decoder. The transformer and convolutional layers extract the high-level global and local fine features from $Y_s$.

- **Decoder:** The decoder consists of three decoding groups corresponding to the encoder, which are used to fuse multi-scale features and restore images. Each decoding group consists of (1) an up-sampling layer to reduce the channel capacity by half and double the size of the feature map, (2) a skip connection layer to fuse deep and shallow features through channel-wise concatenation operation, (3) a $1 \times 1$ convolutional layer to halve the channel number of feature map output by the skip connection layer, and (4) a transformer layer to capture the fine global features. The decoder's feature map $Y_d$ output is fed into a convolutional layer, and the output features are added to a noisy image $Y$ to generate its despeckled counterpart $Y_G$.

Figure 2 illustrates the structure of each transformer block, which is the same as that in Restormer [25]. Generally, a transformer block comprises a Multi-Dconv head Transposed Attention (MDTA) module and a Gated-Dconv Feed-Forward Network (GDFN) module. The MDTA module first generates query (Q), key (K), and value (V) projections enriched by

the local context. Next, it reshapes the query and key projections so that their dot–product interaction generates a transposed-attention map $A$ of size $C \times C$ instead of a huge regular attention map of size $HW \times HW$. Therefore, compared with the original transformer model, it has a higher processing efficiency for high-resolution images. The GDFN module uses a gating mechanism to improve representation learning. The gating mechanism is formulated as the element-wise product of two parallel paths of linear transformation layers, one of which is activated with the GELU non-linearity function [26]. It controls the information flow through the respective hierarchical levels in the pipeline, thereby allowing each level to focus on the fine details complementary to the other levels.

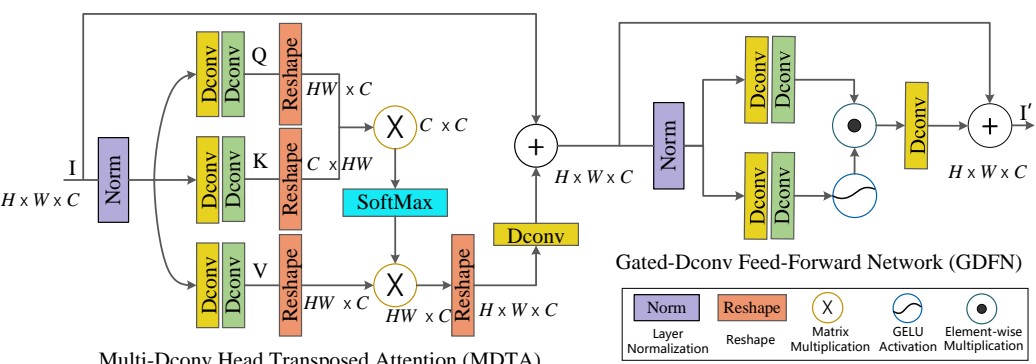

**Figure 2.** The structure of each transformer block [25].

### 3.2.2. The Discriminator Network

For a generated despeckled image $Y_G$ and a clean image $X$, the goal of the discriminator network in SID-TGAN is to distinguish $Y_G$ from $X$. The architecture of the discriminator network adopts that of the Markov discriminator [27] as shown in Figure 3. The discriminator consists of four $3 \times 3$ deep convolutional layers and two transformer layers. The input images first pass through a transformer layer for shallow global feature extraction. Then, deep local features are obtained through four deep convolutional layers. Furthermore, a transformer layer further fuses the multi-scale feature maps. A sigmoid activation function obtains the final discrimination result. After a deep convolution operation in each convolutional layer, a layer normalization operation [28] is adopted to prevent overfitting, and a non-linear activation function LeakyRelu [29] is used to avoid the sawtooth path of gradient updates, which retains the negative gradient information so that it will not be lost entirely.

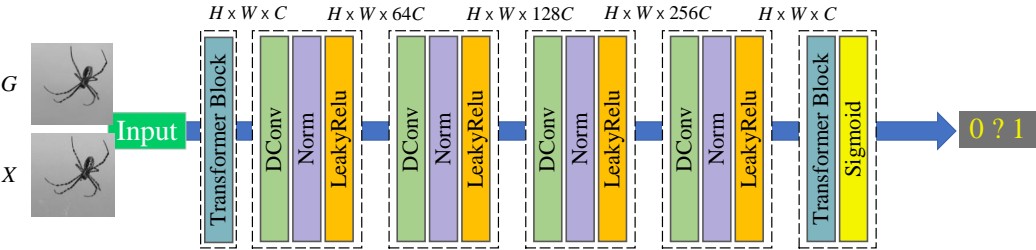

**Figure 3.** The architecture of the discriminator network in SID-TGAN.

### 3.2.3. The Loss Function

Given a source domain $Y$ (of noisy images with speckle noise $N$) and the desired domain $X$ (of clean images), our goal is to learn a mapping $G : \{Y, N\} \rightarrow X$ to perform speckle denoising. The standard GANs are powerful generative models but suffer from instability during training. WGAN-GP [30] penalizes the norm of the gradient of the critic with respect to its input based on the Wasserstein GAN (WGAN) to improve the stability of

the GAN training process and thus generates higher-quality images. Unlike the adversarial loss function of standard GANs in Equation (1), the adversarial loss function of WGAN-GP is expressed as Equation (2):

$$L_{\text{GAN}}(G, D) = \mathbb{E}_{x \sim P_r}[\log D(X)] + \mathbb{E}_{\tilde{x} \sim P_g}[\log(1 - D(\tilde{x}))] \tag{1}$$

$$L_{\text{WGAN-GP}}(G, D) = \mathbb{E}_{\tilde{x} \sim P_g}[D(\tilde{x})] - \mathbb{E}_{x \sim P_r}[D(x)] + \lambda \mathbb{E}_{\hat{x} \sim P_{\hat{x}}}\left[(\| \bigtriangledown_{\hat{x}} D(\hat{x}) \|_2 - 1)^2\right] \tag{2}$$

where the generator $G$ aims to minimize the loss function, while the discriminator $D$ aims to maximize it, $P_r$ is the data distribution, $P_g$ is the model distribution implicitly defined by $\tilde{x} = G(Y, N)$, $P_{\hat{x}}$ denotes uniform sampling along straight lines between pairs of points sampled from the data distribution $P_r$ and the generator distribution $P_g$, and $\lambda$ is a hyperparameter setting of 10 as in [30] by default.

WGAN-GP can only prompt the network to generate an image similar to the reference image, but the processing granularity of the content and global details of the image are relatively coarse. Therefore, we propose a new loss function for SID-TGAN, which contains three additional factors to quantify the loss between the despeckled image and the clean image to guide SID-TGAN to generate higher-quality images. These factors are global similarity, image content, and local texture and style.

**Global Similarity.** Global similarity measures the similarity between two images from a global perspective to detect the degree of image distortion. Since $L_1$ losses are less prone to blurring, the $L_1$ loss in Equation (3) is adopted to calculate the average distance per pixel between the reference and generated images. We simplify the multiplicative speckle noise model as an additive model through a logarithmic transformation to better retain the original information while removing noise. Therefore, the distance between each pixel pair after performing a logarithmic transformation is calculated as follows:

$$L_{\text{GS}}(G) = \mathbb{E}_{X,Y}[\| \log(X) - \log(G(Y)) \|_1] \tag{3}$$

**Image Content**. Image content losses can enhance the ability to describe and distinguish image details and edge information. The $L_2$ loss in Equation (4) is used to measure the content loss between the generated image and the clean image to encourage the generator $G$ to generate an image close to the content of the reference image:

$$L_{\text{IC}}(G) = \mathbb{E}_{X,Y}[\| X - G(Y) \|_2] \tag{4}$$

**Local Texture and Style**. Local textures and styles describe the surface properties of the scene corresponding to an image or image area and preserve the authenticity of the image. We rely on the discriminator $D$ to enforce the consistency of local textures and styles. The adversarial loss function is expressed in Equation (2).

To sum up, the final loss function of SID-TGAN is defined as follows:

$$L = L_{\text{WGAN-GP}} + \alpha L_{\text{GS}} + \beta L_{\text{IC}} \tag{5}$$

where $\alpha$ and $\beta$ are the two hyperparameters indicating the weights of global similarity losses and image content losses.

**Comparison with Prior Art.** SID-TGAN is inspired by Restormer [25], a state-of-the-art transformer-based model for image restoration. However, compared to the original Restormer, SID-TGAN has been significantly improved in the following four aspects. First, the generator network of SID-TGAN streamlines Restormer by reducing the refinement stages and halving the number of input channels of the first-level decoder, thereby lowering the model complexity. Also, the feature aggregation layer introduces additional convolution components to better preserve local feature information in sonar images during global feature extraction. Second, the generator network of SID-TGAN also modifies the number of transformer blocks in the encoder and aggregation layers from 4, 4, 6, 8 to 8, 4, 2, 1.

Employing more transformer blocks on large-scale image blocks can better capture global information; meanwhile, using fewer transformer blocks on small-scale image blocks can prevent over-cleaning and avoid image blurriness. Third, SID-TGAN introduces a novel discriminator to further optimize Restormer. The discriminator distinguishes between generated and real data through local feature extraction, thereby guiding the generator to produce more realistic data and enhancing the quality of denoised images. Fourth, SID-TGAN uses a different loss function from Restormer, which comprehensively considers global similarity, image content, and texture style. This further improves the performance of SID-TGAN in sonar image despeckling.

## 4. Experiments

In this section, we evaluate our proposed SID-TGAN model for speckle denoising through extensive experiments. We first introduce our experimental setup in Section 4.1. Then, the evaluation metrics for the quality of speckle denoising with and without reference images are introduced in Section 4.2. Next, the performance of SID-TGAN compared to the state-of-the-art denoising methods is presented in Section 4.3. Finally, we conduct ablation studies on SID-TGAN in Section 4.4.

### 4.1. Experimental Setup

**Dataset.** A clean sonar image $X$ and a noisy sonar image $Y$ should be input into SID-TGAN to train a speckle denoising model. However, clean sonar images are often not available in practice, whereas obtaining clean optical images is much easier. Thus, we adopted a transfer learning approach that first trains SID-TGAN on paired optical images and then tested it on sonar images. We utilized one optical image dataset and four sonar image datasets. For the optical image dataset, we randomly selected 5500 images of various categories from the ImageNet dataset [31]. Each optical color image was transformed into its corresponding grayscale image to serve as a reference image. Subsequently, we introduced synthetic speckle noise by applying additive noise following a Rayleigh distribution with a mean of 0 and a variance of 1, as well as Gaussian additive noise, to the reference images using Equation (6). This allowed us to create the corresponding synthetic speckle noisy images:

$$Y = X \cdot N_{\mathrm{R}}(\sigma(t)) + N_{\mathrm{G}}, \tag{6}$$

where $N_R$ represents the Rayleigh-distributed noise with parameter $\sigma(t)$ denoting its variance and $N_G$ signifies additive Gaussian noise.

For sonar image datasets, the first two are derived from ocean debris data captured by the ARIS Explorer 3000 [32] forward-looking sonar (FLS), the third is a publicly available side-scan sonar (SSS) image dataset from the 2021 CURPC Competition (https://www.curpc.com.cn, accessed on 21 October 2023), and the fourth is obtained from the SASSED Synthetic Aperture Sonar (SAS) dataset (https://github.com/isaacgerg/synthetic_aperture_sonar_autofocus, accessed on 21 October 2023). Detailed information on the five datasets is provided in Table 1. In the speckle removal task, we randomly selected 4500 optical images as the training set. For the test sets, we chose 500 optical images, 500 forward-looking sonar images, 500 side-scan sonar images, and 100 synthetic aperture sonar images. To further assess the speckle removal performance of different methods, we employed 1457 noisy (or denoised) forward-looking sonar images for training and 111 noisy (or denoised) forward-looking sonar images for testing in the context of target detection. Due to the lack of ground truth, we did not test the side-scan and synthetic aperture sonar images in the target detection task.

**Table 1.** Statistics of datasets in the experiments.

| Dataset | Training Set | Test Set | Is Paired |
|---|---|---|---|
| Optical-Despeckling | 4500 | 500 | Paired |
| FLS-Despeckling | — | 500 | Unpaired |
| SSS-Despeckling | — | 500 | Unpaired |
| SAS-Despeckling | — | 100 | Unpaired |
| Target-Detection | 1457 | 411 | — |

**Methods and Implementation.** The following despeckling methods are compared in our experiments.

- **BM3D** [4]: A filtering-based image denoising algorithm based on the Nash equilibrium balance to decouple the denoising and deblurring operations.
- **BM3D-G** [18]: A filtering-based speckle reduction algorithm that adjusts the parameters of the BM3D algorithm and introduces Gaussian filtering and grayscale correction before basic estimation.
- **NBR2NBR** [13]: A self-supervised framework to train CNN denoisers based only on noisy images.
- **SAR-CNN** [10]: A CNN-based model for SAR image despeckling.
- **SAR-Transformer** [24]: A transformer-based despeckling method with a transformer-based encoder to learn global dependencies between different image regions.
- **SID-TGAN:** Our transformer-based GAN model for sonar image despeckling.

We implemented all models in Python 3 with PyTorch. For a fair comparison, we trained all baseline models in the same configuration, using the source code released by their authors. The hyper-parameters we used include (1) the initial learning rate $3 \times 10^{-4}$, (2) the batch size 4, (3) the number of pre-train epochs from 1 to 150, and (4) the weights of loss functions $\alpha = 0.5$ and $\beta = 1$. In addition, Adam was used as the default optimizer. Data augmentation was performed by flipping horizontally and vertically at random.

All experiments were carried out on a desktop with an Intel® Core™ i5-9600KF CPU @ 3.70 GHz, 16 GB DDR4 RAM, and a Nvidia® GeForce® RTX™ 3090SUPER GPU with 24 GB GDDR6 RAM.

*4.2. Evaluation Criteria*

**Metrics for Despeckling Quality with Reference Images**. When clean reference images and noisy images are both available and have been paired, we use the Peak Signal-to-Noise Ratio (PSNR), Structural Similarity Index Measure (SSIM), and Coincidence degree of the Gray Histogram (GHC) to evaluate the despeckling performance of different methods.

PSNR is an engineering term that represents the ratio of the maximum possible power of a signal to the destructive noise power that affects its representation accuracy. A higher PSNR value represents that the despeckled image is closer to the clean image, thus implying better quality. The calculation of PSNR is presented in Equation (7):

$$PSNR = 10 \log_{10} \left( \frac{(2^n - 1)^2}{MSE} \right), \tag{7}$$

where $MSE$ is the mean square error between the clean image $X$ and the despeckled image $Y_G$, and $n$ is the width of the image $X$.

SSIM measures the structural similarity between the clean images $X$ and the despeckled image $Y_G$. Unlike PSNR, SSIM considers not only the brightness information but also the contrast and structural information of the image, thus having higher accuracy and reliability in evaluating image quality. A larger SSIM value implies a higher similarity between two images. The calculation of SSIM is shown in Equation (8):

$$SSIM(X, Y_G) = [l(X, Y_G)]^{\alpha} [c(X, Y_G)]^{\beta} [s(X, Y_G)]^{\gamma}, \tag{8}$$

and

$$l(X, Y_G) = \frac{2\mu_X \mu_{Y_G} + C_1}{\mu_X^2 + \mu_{Y_G}^2 + C_1}, \tag{9}$$

$$c(X, Y_G) = \frac{2\sigma_X \sigma_{Y_G} + C_2}{\sigma_X^2 + \sigma_{Y_G}^2 + C_2}, \tag{10}$$

$$s(X, Y_G) = \frac{\sigma_{XY_G} + C_3}{\sigma_X \sigma_{Y_G} + C_3}, \tag{11}$$

where $l(X, Y_G)$, $c(X, Y_G)$, and $s(X, Y_G)$ measure the brightness, contrast, and structure of $X$ and $Y_G$, respectively; $\alpha$, $\beta$, and $\gamma$ are the weights to adjust the relative importance of $l(X, Y_G)$, $c(X, Y_G)$, and $s(X, Y_G)$; $\mu_X$ and $\mu_{Y_G}$, and $\sigma_X$ and $\sigma_{Y_G}$ are the mean and standard deviations of $X$ and $Y_G$, respectively; $\sigma_{XY_G}$ is the covariance of $X$ and $Y_G$; and $C_1$, $C_2$, and $C_3$ are all constants to maintain the stability of $l(X, Y_G)$, $c(X, Y_G)$, and $s(X, Y_G)$.

GHC calculates the coincidence rate of the gray pixel value distribution of two images. When the gray distribution of the two images completely overlaps, the value of GHC is 1. Equation (12) presents how GHC is computed:

$$GHC(X, Y_G) = \sum_{i=0}^{n} \begin{cases} 1 - \dfrac{|y(i) - x(i)|}{\max(y(i), x(i))}, & y(i) \neq x(i) \\ 1, & y(i) = x(i) \end{cases} \tag{12}$$

where $y(i)$ and $x(i)$ represent the grayscale pixel distribution of the despeckled image $Y_G$ and the reference image $X$, and $n$ is the range of the pixel values.

**Metrics for Despeckling Quality without Reference**. When reference images were not available, we used the Equivalent Number of Looks (ENL), Speckle Suppression Index (SSI), and Speckle suppression and Mean Preservation Index (SMPI) as metrics for the quality of the despeckling.

ENL measures the smoothness of a uniform area and reflects the ability to remove speckle noise. A higher ENL value represents a higher smoothing efficiency of the speckle noise in homogeneous areas. The calculation of ENL is presented in Equation (13):

$$ENL = \frac{\mu^2}{\sigma^2}, \tag{13}$$

where $\mu$ is the mean value of the image, and $\sigma$ is the standard deviation of the image. Generally speaking, the mean of an image represents its information, while its standard deviation represents its noise severity.

SSI evaluates the speckle noise suppression effect of the denoising model by comparing the mean and standard deviation of pixel values between the noisy image $Y$ and the denoised image $Y_G$. The smaller the SSI value, the better the speckle denoising performance. SSI is defined in Equation (14):

$$SSI = \frac{\sqrt{\sigma(Y_G)}}{\mu(Y_G)} / \frac{\sqrt{\sigma(Y)}}{\mu(Y)} \tag{14}$$

Compared with ENL and SSI, SMPI considers the difference in the mean value between the despeckled image $Y_G$ and the noisy image $Y$. When the mean value of the despeckled image deviates too much from the noisy image, the reliability of the SMPI value is higher than that of ENL and SSI. A lower SMPI value indicates better model performance in terms of mean preservation and noise reduction. SMPI is defined in Equation (15):

$$SMPI = Q \times \frac{\sqrt{\sigma(Y_G)}}{\sqrt{\sigma(Y)}}, \quad \text{where } Q = 1 + |\mu(Y) - \mu(Y_G)| \tag{15}$$

**Metrics for Accuracy of Target Detection**. In addition, we further evaluate the accuracy of target detection on despeckled image output by different methods using three metrics, named mAP, mAP@50, and mAP@75. mAP is the mean of average precision (AP) of detecting all target categories. AP is expressed as the area under the PR (precision and recall) curve from Equation (16):

$$AP = \int_0^1 P(R)dR, \tag{16}$$

where *P* represents the proportion of correctly predicted positive samples among all the predicted positive samples, and *R* represents the proportion of correctly predicted positive samples among all the correctly predicted samples. A higher mAP value represents a higher despeckling quality. mAP50 is the mean of average precision calculated at the IoU (Intersection over Union) threshold value of 0.5. mAP75 is the mean of average precision calculated at the IoU threshold value of 0.75. The mean of the average precision calculated at the IoU thresholds from 0.5 to 0.95 with an interval of 0.05 is called mAP.

*4.3. Performance Evaluation*

**Despeckling Quality for Optical Images with Reference Images.** We compare SID-TGAN with five baselines for their performance on the Optical-Despeckling dataset. The despeckling performance of the six methods is shown in Table 2.

**Table 2.** Performance of different methods on Optical-Despeckling. Here, "↑" means that a higher value indicates better performance on the measure.

| Method | PSNR (↑) | SSIM (↑) | GHC (↑) |
|---|---|---|---|
| BM3D [4] | 19.962 | 0.492 | 0.565 |
| BM3D-G [18] | 18.679 | 0.522 | 0.507 |
| NBR2NBR [13] | 20.810 | 0.631 | 0.542 |
| SAR-CNN [10] | 24.636 | 0.643 | 0.738 |
| SAR-Transformer [24] | 25.203 | 0.671 | 0.732 |
| SID-TGAN | 25.467 | 0.688 | 0.751 |

From Table 2, we observe that SID-TGAN achieves the best despeckling performance in all metrics on optical images. SID-TGAN improves on the existing methods in terms of PSNR, SSIM, and GHC by at least 0.326, 0.020, and 0.013, and at most 6.788, 0.196, and 0.244. SAR-Transformer and SID-TGAN outperform the other four baselines in the PSNR metric, indicating that transformer blocks better capture the global dependencies between different image regions, mitigate the shortcomings of CNNs, and improve the despeckling performance. Unlike SAR-Transformer, SID-TGAN adopts the architecture of GAN and the adversarial loss to generate higher-quality images. Both SID-TGAN and SAR-CNN use discriminative model learning for image despeckling. They achieve the best and second-best GHC values, indicating that the discriminative model drives the local texture and style toward consistency in an adversarial fashion. In addition, we also provide a comparison of the coincidence of the grayscale histogram between the despeckled image returned by each method and the clean image in Figure 4. We can see that the grayscale histogram of the despeckled image returned by SID-TGAN highly overlaps with the grayscale histogram of the clean image. The despeckling performance of SID-TGAN significantly outperforms all other baselines in all evaluation metrics.

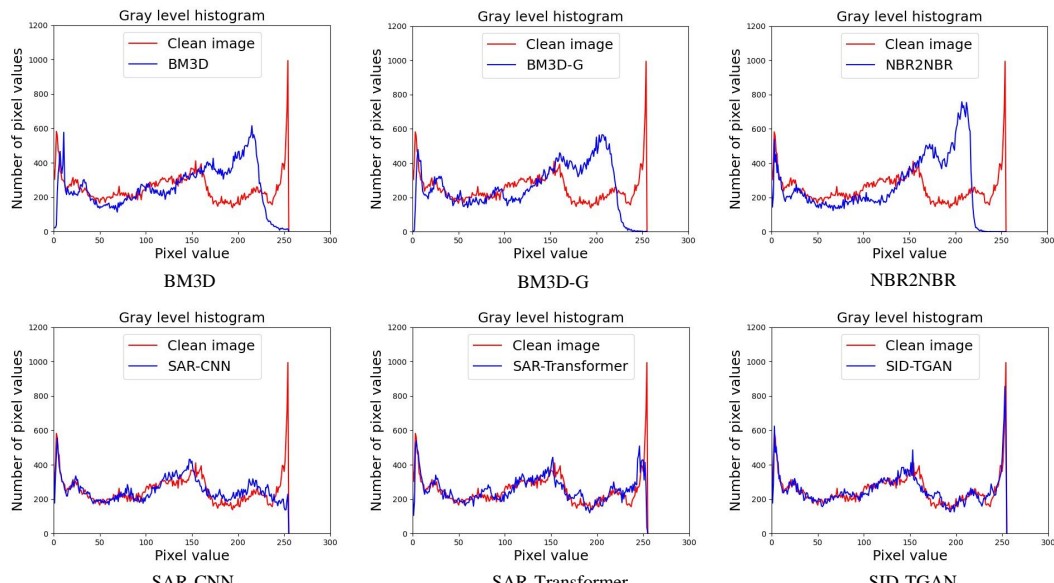

**Figure 4.** Grayscale histogram coincidence between the despeckled image returned by each method and the clean image.

We further visualize the despeckled optical images returned by different methods in Figure 5. The despeckled images generated by SID-TGAN are also closer to clean images from a sensory perspective.

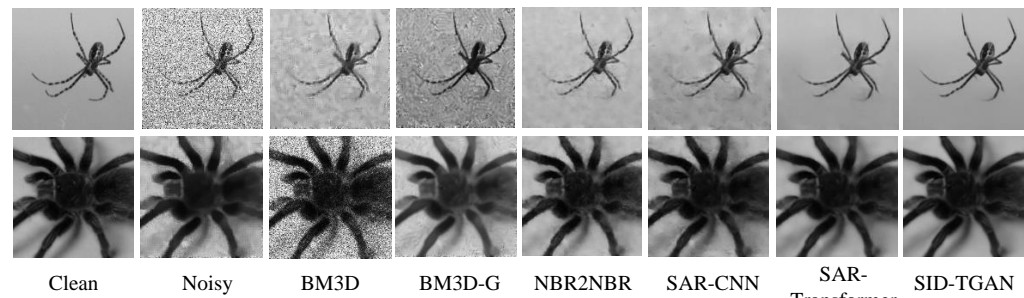

**Figure 5.** Visual comparison of different despeckling methods on optical images.

**Despeckling Quality for Sonar Images without References.** We compare SID-TGAN with five baseline methods on the despeckling performance in the forward-looking sonar (FLS), side-scan sonar (SSS), and synthetic aperture sonar (SAS) image datasets. For each input sonar image with a size of $256 \times 256$, we adopt six different methods to generate the despeckled images. Then, a homogeneous area with a size of $20 \times 20$ and the smallest variance from the noisy sonar image is selected as a homogeneous image patch. The same homogeneous area from six different despeckled images is selected to evaluate the despeckling performance for each method in terms of three quality metrics (i.e., ENL, SSI, and SMPI). The results are shown in Tables 3–5. We also compute the mean standard deviation (MSD) of the homogeneous area from each selected despeckled image patch in Tables 3–5.

**Table 3.** Performance of different methods on FLS-Despeckling. Here, "↑" and "↓" mean that higher and lower values indicate better performance on the measure, respectively.

| Method | ENL (↑) | SSI (↓) | SMPI (↓) | MSD (↓) |
|---|---|---|---|---|
| BM3D [4] | 2053.424 | 0.122 | 0.208 | 1.922 |
| BM3D-G [18] | 1606.703 | 0.133 | 11.305 | 3.412 |
| NBR2NBR [13] | 1731.848 | 0.130 | 0.922 | 1.875 |
| SAR-CNN [10] | 1277.364 | 0.157 | 3.206 | 1.618 |
| SAR-Transformer [24] | 5016.373 | 0.078 | 1.100 | 0.915 |
| SID-TGAN | 9277.537 | 0.063 | 0.572 | 0.715 |

**Table 4.** Performance of different methods on SSS-Despeckling. Here, "↑" and "↓" mean that higher and lower values indicate better performance on the measure, respectively.

| Method | ENL (↑) | SSI (↓) | SMPI (↓) | MSD (↓) |
|---|---|---|---|---|
| BM3D [4] | 1977.021 | 0.213 | 0.213 | 0.022 |
| BM3D-G [18] | 2471.611 | 0.193 | 0.229 | 0.025 |
| NBR2NBR [13] | 2686.033 | 0.154 | 0.150 | 0.013 |
| SAR-CNN [10] | 2837.097 | 0.128 | 0.110 | 0.011 |
| SAR-Transformer [24] | 2443.385 | 0.100 | 0.092 | 0.008 |
| SID-TGAN | 5284.768 | 0.083 | 0.073 | 0.007 |

**Table 5.** Performance of different methods on SAS-Despeckling. Here, "↑" and "↓" mean that higher and lower values indicate better performance on the measure, respectively.

| Method | ENL (↑) | SSI (↓) | SMPI (↓) | MSD (↓) |
|---|---|---|---|---|
| BM3D [4] | 8.489 | 0.680 | 0.459 | 0.008 |
| BM3D-G [18] | 142.902 | 0.124 | 5.202 | 0.517 |
| NBR2NBR [13] | 2.577 | 0.712 | 0.726 | 0.013 |
| SAR-CNN [10] | 2.729 | 0.713 | 0.769 | 0.010 |
| SAR-Transformer [24] | 2.751 | 0.712 | 1.100 | 0.011 |
| SID-TGAN | 3.126 | 0.603 | 0.518 | 0.007 |

Based on the denoising results of each method on the FLS images, as shown in Table 3, SID-TGAN and SAR-Transformer achieve the best and second-best results on the ENL, SSI, and MSD values, once again showing that transformer-based despeckling methods are superior to CNN-based methods. The transformer and CNN blocks of SID-TGAN capture the global and local dependencies between pixels, respectively. Thus, it generates fine despeckled images of the highest quality using the proposed loss function. SID-TGAN is superior to the second-best result by a large margin, with increases of 29% and 34% in terms of SSI and MSD. SID-TGAN is slightly inferior to BM3D in terms of SMPI since SMPI is highly correlated with the size of the selected homogeneous area.

As depicted in Table 4, SID-TGAN demonstrates outstanding speckle suppression performance on SSS images. Its performance notably surpasses that of other comparative methods. Across all four evaluation metrics, SID-TGAN achieves the best results and, compared to the second-best method, shows performance improvements of 17%, 21%, and 13% in terms of SSI, SMPI, and MSD, respectively. These results underscore the exceptional applicability of SID-TGAN in the context of speckle reduction for SSS images.

Upon scrutinizing the results in Table 5, we find that SID-TGAN demonstrates commendable denoising capabilities on SAS images as well. Notably, our chosen SAS dataset predominantly exhibits images with lower overall brightness. BM3D-G [18] involves gamma correction, a preprocessing step that adjusts the brightness and contrast of input images. This adjustment has a substantial influence on the calculation of the ENL value in the evaluation, which ultimately results in considerably higher ENL values for the output images produced by BM3D-G that surpass those of any other method. We also note that the noise distribution in SAS images differs markedly from those of FLS and SSS images.

However, the training set employed in the experiment is tailored to simulate the speckle noise distribution from FLS and SSS images by generating synthetically matched pairs from optical grayscale images. Consequently, the model parameters are not ideally suited for SAS images, which leads to the degraded performance of SID-TGAN. In contrast, BM3D [4] does not depend on the types of noise distribution and outperforms SID-TGAN. Nevertheless, we observe that SID-TGAN still performs much better than other deep learning-based methods on SAS images.

We present the visualizations of the despeckled sonar images returned by different methods in Figure 6. The first row depicts the despeckled sonar image of size $256 \times 256$, while the second row portrays the despeckled homogeneous area image patch, sized $20 \times 20$ extracted from the red box in the first row of Figure 6. In summary, SID-TGAN outperforms all other methods in terms of quantitative and sensory analysis on forward-looking sonar images.

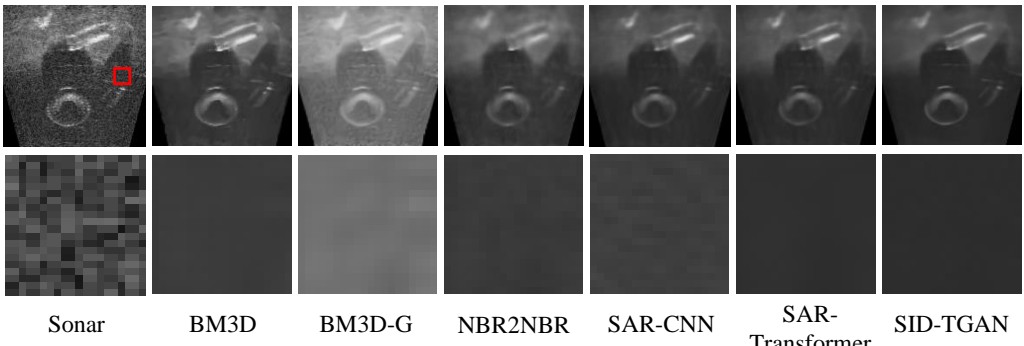

**Figure 6.** Visual comparison of different despeckling methods on forward-looking sonar images. The red frame in the first image indicates the position of the despeckled homogeneous area image patch presented in the second row.

**Accuracy for Target Detection on Despeckled Sonar Images**. We compare the accuracy of target detection using Faster-RCNN [33] on noisy and despeckled sonar images to evaluate the improvements of different despeckling methods in image quality. The results for target detection on the sonar-detection dataset are shown in Table 6. Undoubtedly, despeckled sonar images provided by SID-TGAN and SAR-Transformer [24] exhibit superior image quality. Compared to noisy sonar images, despeckled sonar images provided by SID-TGAN show a remarkable improvement in all three accuracy metrics for target detection, i.e., mAP, mAP@50, and mAP@75 by 4.5%, 1.7%, and 8.4%, respectively. It is surprising that the mAP, mAP@50, and mAP@75 of the despeckled sonar images returned by NBR2NBR [13] and SAR-CNN [10] are even lower than those of noisy sonar images. This implies that NBR2NBR [13] and SAR-CNN [10] fail to effectively remove the speckle noise in the sonar images. After despeckling sonar images with two filtering-based methods, namely BM3D [4] and BM3D-G [18], the mAP, mAP@50, and mAP@75 metrics show marginal improvements but are still inferior to transformer-based despeckling methods, i.e., SID-TGAN and SAR-Transformer [24]. We also visualize the effects of using despeckled images by different methods for the target detection task in Figure 7. The above results confirm that SID-TGAN achieves better despeckling performance and provides sonar images of higher quality for target detection compared to existing despeckling methods.

**Table 6.** Target detection accuracy on sonar images using different despeckling methods. Here, "↑" means that a higher value indicates better performance on the measure.

| Method | mAP (↑) | mAP@50 (↑) | mAP@75 (↑) |
|---|---|---|---|
| No-despeckling | 0.584 | 0.937 | 0.631 |
| BM3D [4] | 0.513 | 0.874 | 0.528 |
| BM3D-G [18] | 0.569 | 0.933 | 0.596 |
| NBR2NBR [13] | 0.540 | 0.904 | 0.581 |
| SAR-CNN [10] | 0.602 | 0.946 | 0.678 |
| SAR-Transformer [24] | 0.605 | 0.947 | 0.681 |
| SID-TGAN | 0.610 | 0.953 | 0.698 |

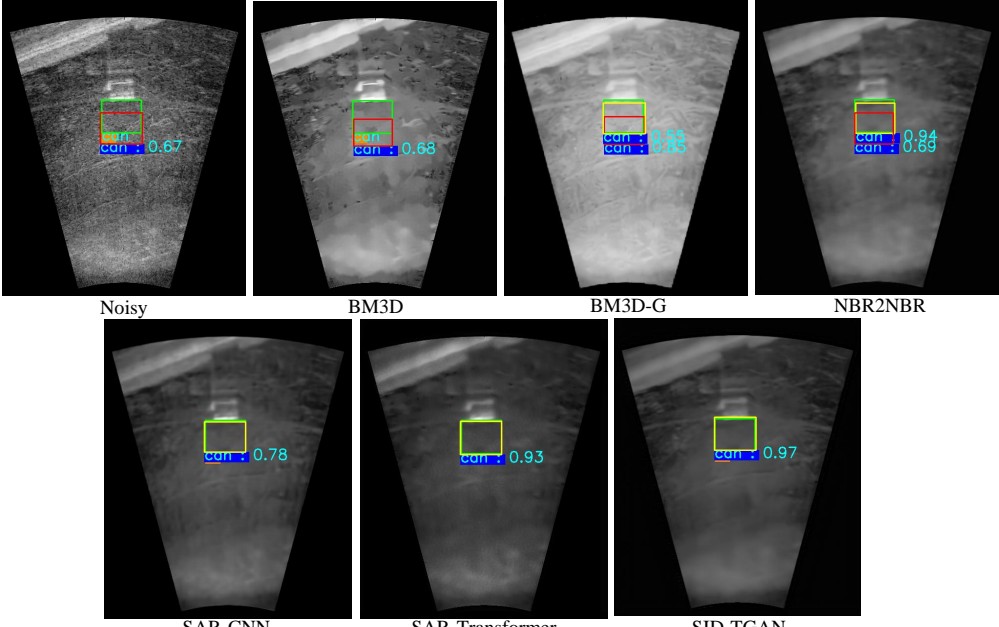

**Figure 7.** Visual comparison of target detection on different despeckled sonar images. In each image, the green box indicates the ground-truth target position, the red box indicates a wrong predicted target position due to noise, the yellow box indicates a correct predicted target position owing to denoising, and the numbers indicate the probabilities that the target object is classified as its true category, i.e., "can".

**Efficiency Evaluation**. We present the running time required by each method to train the model and to perform the denoising process on a single image in Table 7. It should be noted that SID-TGAN, in comparison to CNN-based denoising methods, exhibits slightly higher training and denoising time. This can be attributed to the fact that the transformer model needs to extract global positional information when processing input data, which, in contrast to the localized nature of convolutional operations, results in prolonged processing time. In addition, filtering-based approaches do not require any training procedure in advance but take more than three orders of magnitude longer time in the denoising process.

**Table 7.** The time required by each method to train the model and to perform the denoising process on a single image.

| Method | Training Time (h) | Denoising Time (s) |
|---|---|---|
| BM3D [4] | — | 82.59 |
| BM3D-G [18] | — | 81.42 |
| NBR2NBR [13] | 7.58 | 0.0156 |
| SAR-CNN [10] | 3.61 | 0.0157 |
| SAR-Transformer [24] | 13.49 | 0.0161 |
| SID-TGAN | 24.01 | 0.0685 |

### 4.4. Ablation Studies

The despeckling performance of SID-TGAN is improved through the following key contributions. First, it integrates the transformer layer into the GAN framework to capture global dependencies among image pixels. Second, it optimizes Restormer [25] by removing the final refinement stage and introducing deep convolutional layers in the feature integration component to improve the extraction of global features while retaining local features. Third, it proposes a new loss function that promotes the generation of high-quality images in SID-TGAN by considering global similarity, image content, and local texture and style. To evaluate the impact of each factor on the despeckling performance of SID-TGAN, we conduct ablation studies on the optical-despeckling and sonar-despeckling datasets. We utilize two widely adopted full reference image quality metrics, PSNR and SSIM, to evaluate the despeckling performance on the optical images. Similarly, two image quality metrics without references, SSI and SMPI, are employed to evaluate the despeckling performance on the sonar images.

**Effect of Transformer Module.** We compare the denoising performance of SID-TGAN with and without the transformer layer on FLS, SSS, and SAS images in Tables 8 and 9. Another baseline is the original WGAN [30] model with weight clipping and gradient penalty. From Table 8, we can see that adding transformer layers to the discriminator of SID-TGAN marginally improves its performance, while adding transformer layers to the generator of SID-TGAN improves its performance much more significantly. After adding transformer layers to the generator of SID-TGAN, the values of PSNR, SSIM, SSI, and SMPI increase by 15%, 27%, 31%, and 78%, respectively. When transformer layers are added to both the generator and discriminator networks, SID-TGAN achieves the best despeckling performance. Table 9 reveals that the incorporation of the transformer module into SID-TGAN leads to significant improvements in the denoising performance for SSS and SAS images as well. For SSS images, the SSI and SMPI values increase by 93% and 94%. For SAS images, the SSI and SMPI values also show improvements of 58% and 89%. The above results indicate that the transformer module effectively captures global dependencies between pixels, compensating for the limitations of convolutional operations that only capture local dependencies between pixels, thus significantly improving the despeckling performance of SID-TGAN.

**Table 8.** Performance of SID-TGAN with or without the transformer module on Optical-Despeckling and FLS-Despeckling, where *D* and *G* are the abbreviations for the discriminator and generator of SID-TGAN. Here, "↑" and "↓" mean that higher and lower values indicate better performance on the measure, respectively.

| Model | Transformer | PSNR (↑) | SSIM (↑) | SSI $_{FLS}$ (↓) | SMPI $_{FLS}$ (↓) |
|---|---|---|---|---|---|
| WGAN [30] | w/o | 22.062 | 0.540 | 0.350 | 4.247 |
| SID-TGAN | wD w/oG | 22.258 | 0.544 | 0.346 | 4.131 |
| | wG w/oD | 25.441 | 0.673 | 0.067 | 1.003 |
| | wG wD | 25.467 | 0.688 | 0.063 | 0.572 |

**Table 9.** Performance of SID-TGAN with or without the transformer module on SSS-Despeckling and SAS-Despeckling, where *D* and *G* are the abbreviations for the discriminator and generator of SID-TGAN. Here, "↓" means that a lower value indicates better performance on the measure.

| Model | Transformer | $\text{SSI}_{\text{SSS}}(\downarrow)$ | $\text{SMPI}_{\text{SSS}}(\downarrow)$ | $\text{SSI}_{\text{SAS}}(\downarrow)$ | $\text{SMPI}_{\text{SAS}}(\downarrow)$ |
|---|---|---|---|---|---|
| WGAN [30] | w/o | 1.168 | 1.151 | 1.434 | 4.678 |
| SID-TGAN | wD w/oG | 0.769 | 0.952 | 1.337 | 4.060 |
| | wG w/oD | 0.088 | 0.077 | 0.712 | 0.627 |
| | wG wD | 0.083 | 0.073 | 0.603 | 0.518 |

**Improvements upon Restormer's Performance.** We conducted a series of experiments to show how the performance of SID-TGAN is improved upon Restormer to highlight the exceptional capabilities of SID-TGAN. Tables 10 and 11 present the results, where the baseline model is Restormer itself, and ResGAN replaces the generator with Restormer while utilizing the discriminator from SID-TGAN. The results in Tables 10 and 11 demonstrate that by introducing the discriminator into the Restormer model, removing the refinement layers in Restormer, and adding deep convolutional layers in the encoder output stage, as well as incorporating dimensionality reduction operations in the primary decoder, we further extract advanced and refined global and local features. This improvement has had a significant impact on the despeckling performance across three sonar image datasets. On FLS images, the SSI and SMPI values increase by 63% and 76%. On SSS images, the SSI and SMPI values show improvements of 22% each. On the SAS images, the SSI and SMPI values increase by 20% and 13%.

**Table 10.** Performance comparison of different feature integration components, Restormer, ResGAN, and SID-TGAN, on Optical-Despeckling and FLS-Despeckling. Here, "↑" and "↓" mean that higher and lower values indicate better performance on the measure, respectively.

| Feature Integration | PSNR (↑) | SSIM (↑) | $\text{SSI}_{\text{FLS}}(\downarrow)$ | $\text{SMPI}_{\text{FLS}}(\downarrow)$ |
|---|---|---|---|---|
| Restormer [25] | 25.397 | 0.681 | 0.170 | 2.391 |
| ResGAN | 25.459 | 0.686 | 0.079 | 1.220 |
| SID-TGAN | 25.467 | 0.688 | 0.063 | 0.572 |

**Table 11.** Performance comparison of different feature integration components, Restormer, ResGAN, and SID-TGAN, on SSS-Despeckling and SAS-Despeckling. Here, "↓" means that a lower value indicates better performance on the measure.

| Feature Integration | $\text{SSI}_{\text{SSS}}(\downarrow)$ | $\text{SMPI}_{\text{SSS}}(\downarrow)$ | $\text{SSI}_{\text{SAS}}(\downarrow)$ | $\text{SMPI}_{\text{SAS}}(\downarrow)$ |
|---|---|---|---|---|
| Restormer [25] | 0.106 | 0.094 | 0.752 | 0.593 |
| ResGAN | 0.089 | 0.075 | 0.716 | 0.546 |
| SID-TGAN | 0.083 | 0.073 | 0.603 | 0.518 |

**Effect of Loss Function Component.** The loss function of SID-TGAN consists of the global similarity loss $L_{\text{GS}}$, the image content loss $L_{\text{IC}}$, and the local texture and style loss $L_{\text{WGAN-GP}}$. We evaluate the impact of each component of the loss function on the despeckling performance of SID-TGAN in Tables 12 and 13. Since SID-TGAN is a GAN model, $L_{\text{WGAN-GP}}$ is the default adversarial loss function. After $L_{\text{GS}}$ or $L_{\text{IC}}$ is added to $L_{\text{WGAN-GP}}$, the despeckling performance of SID-TGAN improves significantly for both optical images and sonar images. These results confirm that the global similarity loss function $L_{\text{GS}}$ better retains global information while removing noise. Similarly, the image content loss function $L_{\text{IC}}$ also effectively improves image quality by better preserving the original information in the image. After both $L_{\text{GS}}$ and $L_{\text{IC}}$ are added to $L_{\text{WGAN-GP}}$, SID-TGAN achieves the best performance.

By examining Tables 12 and 13, we can deduce that the $L_{\text{IC}}$ loss function has a significantly greater impact on the performance of the model compared to the $L_{\text{GS}}$ loss function.

When considering the comprehensive loss in Equation (5) and after extensive experimentation involving parameter adjustments, we observe that when the hyperparameter $\alpha$ is set too small, the contribution of $L_{GS}$ to the total loss becomes marginal, resulting in the retention of certain levels of blur in the denoised images. Conversely, when $\alpha$ is set too high, the prominence of the $L_{GS}$ model's weight interferes with $L_{IC}$'s ability to handle image details and edge information, thus adversely affecting the overall model performance. As a result, based on our experimental findings, we accordingly configured the values of $\alpha$ and $\beta$ as 0.5 and 1, respectively, where the model achieves the best overall performance.

**Table 12.** Performance of SID-TGAN on Optical-Despeckling and FLS-Despeckling with different loss functions. Here, "↑" and "↓" mean that higher and lower values indicate better performance on the measure, respectively.

| Loss Function | PSNR (↑) | SSIM (↑) | $\text{SSI}_{\text{FLS}}$ (↓) | $\text{SMPI}_{\text{FLS}}$ (↓) |
|---|---|---|---|---|
| $L_{\text{WGAN-GP}}$ | 13.346 | 0.215 | 1.179 | 12.169 |
| $L_{\text{WGAN-GP}} + L_{\text{GS}}$ | 25.421 | 0.681 | 0.067 | 0.972 |
| $L_{\text{WGAN-GP}} + L_{\text{IC}}$ | 25.435 | 0.683 | 0.082 | 1.263 |
| $L_{\text{WGAN-GP}} + L_{\text{IC}} + L_{\text{GS}}$ | 25.467 | 0.688 | 0.063 | 0.572 |

**Table 13.** Performance of SID-TGAN on SSS-Despeckling and SAS-Despeckling with different loss functions. Here, "↓" means that a lower value indicates better performance on the measure.

| Loss Function | $\text{SSI}_{\text{SSS}}$ (↓) | $\text{SMPI}_{\text{SSS}}$ (↓) | $\text{SSI}_{\text{SAS}}$ (↓) | $\text{SMPI}_{\text{SAS}}$ (↓) |
|---|---|---|---|---|
| $L_{\text{WGAN-GP}}$ | 1.100 | 1.022 | 1.304 | 1.965 |
| $L_{\text{WGAN-GP}} + L_{\text{GS}}$ | 0.139 | 0.128 | 0.698 | 0.536 |
| $L_{\text{WGAN-GP}} + L_{\text{IC}}$ | 0.103 | 0.094 | 0.699 | 0.524 |
| $L_{\text{WGAN-GP}} + L_{\text{IC}} + L_{\text{GS}}$ | 0.083 | 0.073 | 0.603 | 0.518 |

## 5. Conclusions and Future Work

This paper introduces SID-TGAN, a sonar image despeckling model based on generative adversarial networks. By integrating transformer blocks for global feature extraction and convolutional blocks for local (spatial) feature extraction within the generator and discriminator networks, SID-TGAN effectively extracts and enhances useful features from sonar images. SID-TGAN comprehensively learns features from the training data by leveraging adversarial training, resulting in better despeckling performance. Furthermore, by adopting a novel adversarial loss function, SID-TGAN emphasizes the overall consistency of images and ensures the meticulous representation of local features, leading to significant improvements in image quality with more intricate detail preserved. Extensive experimental results on synthetic optical noise image datasets and real sonar image datasets demonstrate that SID-TGAN significantly outperforms state-of-the-art filtering methods and CNN-based despeckling approaches in terms of despeckling performance. In general, our proposed SID-TGAN model effectively reduces speckle noise in sonar images, offering a promising solution to improve image quality and preserve important features in subsequent sonar image analysis tasks.

For future work, we would like to delve into the application of SID-TGAN in other sonar image-processing tasks and further optimize its performance in diverse noise patterns and complex scenarios. This endeavor is poised to bring forth novel insights and breakthroughs in the realm of sonar image-processing research.

**Author Contributions:** X.Z.: Conceptualization, Formal analysis, Investigation, Writing—original draft; K.T.: Data curation, Methodology, Software, Writing—review and editing; Z.Z.: Investigation, Resources, Visualization; B.N.: Funding acquisition, Project administration, Supervision; Y.W.: Validation, Writing—review and editing. All authors have read and agreed to the published version of the manuscript.

**Funding:** This work was supported by the National Natural Science Foundation of China under grant nos. 61976032 and 62002039.

**Data Availability Statement:** The data that support the findings of this study are available from the corresponding author upon reasonable request.

**Conflicts of Interest:** The authors declare that they have no known competing financial interest or personal relationships that could have appeared to influence the work reported in this paper.

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
