# Peer review of "SID-TGAN: A Transformer-Based Generative Adversarial Network for Sonar Image Despeckling"

_remotesensing, doi:10.3390/rs15205072_

Round 1

Reviewer 1 Report

The author proposed a novel transformer-based generative adversarial network named SID-TGAN for sonar image despeckling. By integrating transformer blocks for global feature extraction and convolutional blocks for local (spatial) feature extraction within the generator and discriminator networks, SID-TGAN effectively extracts and enhances useful features from sonar images.

1. This manuscript mentioned many side scan sonar (SSS) image processing methods. Can the proposed method be applied to SSS image processing? Please add a related experiment.

2. Please give the time cost of each method mentioned in the experiment part. 

3. In Fig.7, it seems that the rectangles did not cover the target, why? Isn't this a object detetion epxeriment?Please explain.

Reviewer 2 Report

In this paper, the authors propose a novel transformer-based generative adversarial network named SID-TGAN for sonar image despeckling. In general, the work in this paper is very important for the improvement of sonar image. However, the revisions should be carried out before acceptance.

1. In practice, the synthetic aperture sonar image [1][2] would often suffer from speckle noise, as this sonar is based on the coherent signal processing. That is to say, the authors should further discuss the despeckling methods about synthetic aperture sonar. The reference review should be enhanced.

[1] Zhang. An omega-k algorithm for multireceiver synthetic aperture sonar. Electronics Letters. 2023, Doi: 10.1049/ell2.12859

[2]Yang. An imaging algorithm for high-resolution imaging sonar system. Multimedia Tools and Applications. 2023, Doi: 10.1007/s11042-023-16757-0

2. The references in this paper are mostly conference papers. The authors should cite recent published papers from highly recognized journals like Remote Sensing, IEEE GRSL, Electronics Letters and so on.

3. In line 210, the lamda is chosen as 10. Furthermore, the parameters shown as lamda in Eq. (5) are still used. The reviewer wanders to know how to determine this parameter. The authors should discuss the reason. Besides, the influence of this parameter on performance should be discussed. Further

4. The loss function is discussed in section 3.2.3. However, the robustness of the authors’ method is not discussed. The authors should discuss it in their paper.

5. The synthetic aperture sonar images are strongly suggested to be processed by authors’ method in section 4.

6. The English in this paper should be improved.

Further improvement should be done

Reviewer 3 Report

This paper presents a new application of transformer neural networks to the problem of noise removal from image rasters. In this case, authors present a use case for sonar imagery but use a large training set of paired clean-noisy data from "optical" images, photographs (their Table 1). The sonar data is a test set and training set introduced later in the paper. So, the NN architecture applies to images broader than sonar datasets and is a sophisticated version of a general class of models that use a Generative Adversarial Network (GAN) for modeling noise and removing that model from imagery. My own knowledge of this area is not vast, as a Geoscientist that uses AI and not a Data Scientist, but the model presented seems to be novel enough to warrant publication. It is perhaps not unexpected that the model presented performs better than the alternatives presented in this paper because the GAN are already known to outperform most other methods for "cleaning up" images.  The alternatives, BM3D and a CNN, are older and those are widely used for the sake of simplicity and utility.

What emerges as being important after all of the analysis presented in the tables in this paper is that their model (SID-TRAN) and the other transformer model (SAR-Transformer) perform on par with each other. This leads to the conclusion, which has been published in quite a few papers already but is still a new advance, is that transformers are what is important here. The structure of their network for SID-TRAN appears to be very effective at speckle removal. It's unclear if it is particularly efficient and no data on the run times are given.

I think that the images are harvested from large datasets (lines 262-277) and that the images used should be available to inspect in the supplemental data, or that more detail on how images were selected be provided in the Experimental Setup section.  It is also unclear how the speckle noise was generated from the Rayleigh Distribution and Gaussian additive noise was exactly implemented. Thus, it would be difficult for me to replicate their exact process. 

Although I have no problem with publishing this paper in it's current form, I feel that it's important to note that SID-TRAN is more than a simple despeckling algorithm. It actually performs image enhancement as the transformer learns the useful features in the image. The authors statements in the abstract and Conclusions reveal this fact. It is, in fact, feature enhancement.

Round 2

Reviewer 1 Report

The author has addressed all my concerns.

Reviewer 2 Report

The authors well addressed my comments